# Pet Owners’ Knowledge of Antibiotic Use and Antimicrobial Resistance and Their Antibiotic Practices: Comparison Between Contexts of Self and Pet

**DOI:** 10.3390/antibiotics14020158

**Published:** 2025-02-05

**Authors:** Seema Aithal, Huiling Guo, Boon Han Teo, Timothy Chua, Zoe Jane-Lara Hildon, Angela Chow

**Affiliations:** 1Department of Preventive and Population Medicine, Office of Clinical Epidemiology, Analytics, and Knowledge, Tan Tock Seng Hospital, 11 Jalan Tan Tock Seng, Singapore 308433, Singapore; seema_k_aithal@ttsh.com.sg (S.A.); huiling_guo@ttsh.com.sg (H.G.); 2Singapore Veterinary Association, Singapore 069110, Singapore; teo.boon.han@vettrustsingapore.com; 3Beecroft Animal Specialists & Emergency Hospital, Singapore 119973, Singapore; timothy.chua@beecroft.com.sg; 4Saw Swee Hock School of Public Health, National University of Singapore, Singapore 117459, Singapore; zhildon@nus.edu.sg; 5Lee Kong Chian School of Medicine, Nanyang Technological University, Singapore 308232, Singapore

**Keywords:** One Health, antimicrobial resistance, pet owners, antibiotic practice

## Abstract

**Objective**: This study aimed to compare pet owners’ knowledge of antibiotic use, understanding of antimicrobial resistance (AMR) and antibiotic practices for themselves and their pets to guide behavioral interventions to reduce AMR. **Methods**: A cross-sectional study conducted between March 2023 and December 2023 involved 1080 pet owners recruited from 16 veterinary clinics in Singapore. An anonymous self-administered survey assessed the participants’ knowledge of antibiotic use and AMR as well as their adherence to recommended antibiotic practices for themselves and their pets. McNemar’s Chi-square test was used to identify significant differences in the outcome variables between self and pets. The differences between the type of pet owned (dog owner/non-dog owner) were assessed using Pearson’s Chi-square test. **Results**: Among the 1080 pet owners surveyed, poorer knowledge of antibiotic use (46.3% vs. 33.2%; *p* < 0.001) and inappropriate antibiotic use (33.9% vs. 23.5%; *p* < 0.001) was more common for participants’ pets than for themselves. Compared to dog owners, non-dog owners had poorer knowledge of antibiotic use for their pets (poor knowledge, 55.2%; 42.4%; *p* < 0.001), but the knowledge of AMR occurrence and antibiotic practices did not differ between the dog owners and the non-dog owners. **Conclusions**: Pet owners have significantly poorer antibiotic use knowledge and antibiotic practices for their pets than for themselves. Compared to dog owners, non-dog owners have poorer knowledge of antibiotic use. Educational initiatives addressing pet owners’ knowledge deficits may improve their antibiotic practices for their pets and themselves, emphasizing the importance of cross-sectoral One Health planning.

## 1. Introduction

Antimicrobial resistance (AMR) is a serious public health problem facing humanity. If left unchecked, AMR could lead to as many as 10 million deaths by 2050 [1]. To combat this threat, a “One Health” approach that seeks the collaborative efforts of various disciplines across different sectors to achieve optimal health for humans, animals and their shared environment has been strongly emphasized by the World Health Organization (WHO) [2].

The overuse and misuse of antibiotics promote AMR. The risk of humans acquiring AMR from pets and vice versa is not uncommon owing to their close physical proximity for prolonged durations and because they share a similar reservoir of antibiotic-resistant bacteria [3,4,5,6]. Moreover, the paucity of new antimicrobials for the treatment of multidrug-resistant infections and a substantial overlap between the frequently used antibiotics between humans and pet animals further complicate matters [7,8]. With the rapid increase in pet ownership worldwide [9,10] due to the growing popularity of pets, there is a pressing need to ensure the responsible and judicious use of antibiotics in pets [11].

Prior studies among pet owners [12,13,14,15,16,17,18,19] have found glaring deficiencies in their knowledge and attitudes concerning the appropriate use of antibiotics in pets. Smith et al. reported that inappropriate antimicrobial prescribing may be attributed to veterinarians succumbing to perceived pressure or direct requests from clients [17]. While many of these insights have been garnered from studies in Western countries, there is a dearth of data from an Asian context, where countries already grappling with a high AMR burden [20,21] are reporting steadily growing pet populations due to a burgeoning affluent middle class [22]. To date, most of the AMR literature from Asia has involved laboratory-based studies, comprising the genotyping of bacterial isolates taken from companion animals to analyze antimicrobial resistance patterns [23,24,25,26,27,28].

No studies have previously examined the comparative knowledge, attitudes and practices of pet owners surrounding the use of antibiotics and AMR for themselves and for their pets. A recent qualitative study reported that pet owners applied the antimicrobial information received from their own doctors to their pet’s health care [29]. As such, educating pet owners about appropriate antibiotic use for themselves may lead to improved antibiotic practices for their pets. In addition, there is a growing emphasis on the need to embed behavioral insights into novel strategies targeting inappropriate use, as AMR is primarily driven by behaviors associated with the misuse and overuse of antimicrobials [30,31]. Thus, accruing baseline data on the pet owners’ existing knowledge, perceptions and practices surrounding the use of antibiotics is a vital first step to fill this knowledge gap. This may help to inform future research on the design of effective behavior change interventions to increase the awareness of AMR among pet owners.

In this study, we aimed to compare pet dog and/or cat owners’ knowledge of antibiotic use and AMR and antibiotic practices for themselves and for their pets in Singapore, an Asian country with rising pet ownership [32,33]. Additionally, we assessed the knowledge, attitudes and antibiotic practices by type of pet owned (dog owner/non-dog owner), which will subsequently guide the design and implementation of interventions to reduce the inappropriate use of antibiotics in pets.

## 2. Results

A total of 1933 eligible pet owners (adults aged 21 years and above) of dogs and/or cats were approached at 16 veterinary clinics across Singapore, and 1080 completed responses were collected, with a survey response rate of 56%.

### 2.1. Baseline Characteristics

Of 1080 pet owners, the majority were female (64%), higher educated (87%), currently employed (76%) and living in large housing (65%). Nearly 70% of the pet owners owned dogs (which included those who owned both dog(s) and cat(s)—6%), and 475 (44%) had over 10 years of pet ownership experience. A significantly higher proportion of dog owners were ethnic Chinese (82.4%; 54.3%; *p* < 0.001) and lived in large housing (70.0%; 53.3%; *p* < 0.001). While almost half (47.9%, 361/754) of those who owned dogs had >10 years of pet ownership experience, only one-third (35%, 114/326) of non-dog owners had >10 years of pet ownership experience (*p* < 0.001) (Table 1).

### 2.2. Knowledge of Antibiotic Use

In general, pet owners tended to have poorer knowledge of antibiotic use for their pets than for themselves (46.3% vs. 33.2%; *p* < 0.001), with 38% of pet owners reporting that “It is okay to buy or request the same antibiotics, if they helped my pet get better previously”, but only 28% of pet owners followed this practice for themselves (*p* < 0.001). While 23% of pet owners reported “It is okay to re-use antibiotics given to another pet to treat the same illness”, only 11% of them reported that it was okay for themselves to use antibiotics that had been given to another friend/family member (*p* < 0.001) (Figure 1).

Compared to dog owners, non-dog owners had poorer overall antibiotic knowledge for their pets (poor knowledge, 55.2%; 42.4%; *p* < 0.001), with a higher proportion of non-dog owners reporting that “It is okay to buy or request the same antibiotics, if they helped my pet get better previously” (45% vs. 35%; *p* = 0.002) and “It is okay to re-use antibiotics given to another pet if used to treat the same illness” (29% vs. 21%; *p* = 0.006) [Appendix A].

### 2.3. Knowledge of Antimicrobial Resistance

There was no significant difference between the pet owners’ overall knowledge of AMR occurrence in humans and pets (65% vs. 67%; *p* = 0.13). While 46% of the pet owners were aware that bacteria that are resistant to antibiotics can spread from person to person, only 23–24% were aware that such bacteria can spread from pets to their pet owners and vice versa (*p* < 0.001). Similarly, a higher proportion of pet owners recognized that many infections were becoming increasingly resistant to treatment with antibiotics in humans, but fewer pet owners believed this to be true in pets (67% vs. 38%; *p* < 0.001) [Figure 2]. However, similar proportions of pet owners believed that infections caused by drug-resistant bacteria are difficult to treat in both humans and pets (73% vs. 70%; *p* = 0.03) [Appendix A].

Compared to dog owners, non-dog owners had a similarly poor knowledge of AMR occurrence in pets (70.3%; 65.8%; *p* = 0.152). More dog owners were cognizant of infections becoming increasingly resistant to treatment with antibiotics compared to non-dog owners (40.5% vs. 32%; *p* < 0.001) (Appendix A).

### 2.4. Antibiotic Practices

We observed that pet owners tended to use antibiotics inappropriately for their pets to a greater extent than for themselves (33.9% vs. 23.5%; *p* < 0.001). Regarding antibiotic practices, while 13% of pet owners stopped taking antibiotics when they started feeling better, 19.5% of pet owners stopped giving their pets antibiotics when their pets started feeling better or when their symptoms subsided (*p* < 0.001). Only 5.6% of pet owners reportedly shared their antibiotics with a sick family member, but 9.8% of them gave their antibiotics to their sick pets (*p* < 0.001) (Figure 3)

There was no significant difference in antibiotic practices between dog owners and non-dog owners (poor practice 34.6%; 32.2%; *p* = 0.44) (Appendix A).

## 3. Discussion

We observed that pet owners tended to have poorer knowledge of antibiotic use (46.3% vs. 33.2; *p* < 0.001) and used antibiotics more inappropriately (33.9% vs. 23.5%; *p* < 0.001) for their pets than for themselves. To date, none of the published international literature has made the comparison between pet owners’ knowledge of antibiotic use for their pets and for themselves. Notably, in our study, we found that 23% of pet owners were content with re-using antibiotics that had been given to another pet and prescribed for the same or a similar illness, although only 11% of pet owners reported that it was acceptable for themselves to use antibiotics that had been given to another friend/family member. A prior qualitative study reported that pet owners usually err on the side of caution and administer antibiotics to their pets to prevent the risk of infection [12]. Pet owners’ anxiety over their pets’ health and their inexperience in managing their pets’ illnesses may have led to their poor antibiotic practices. This observation not only underscores the value of a One Health approach in educating pet owners about appropriate antibiotic use but also highlights the need to explore the underlying contexts and mechanisms that drive unnecessary antibiotic use for pets. Despite the recognition of patients (and, by extension, pet owners) as key stakeholders in prescribing decisions, it is still unclear regarding the most effective ways of educating the public about appropriate antibiotic use [34,35]. Wright et al. [36] reported that pet owners displayed a higher awareness of the impact of AMR and were more amenable to antimicrobial stewardship measures after watching a short AMR engagement animation. Such specific behavioral interventions may facilitate pet-owner engagement in improving AMR awareness and reducing inappropriate antimicrobial prescribing.

Our results showed that pet owners’ poor adherence to antibiotic regimens was more prevalent in the treatment for pets compared to themselves (stopped antibiotic treatment when symptoms improved, 19.5% vs. 13%; *p* < 0.001). This concurred with an earlier U.S. study, which reported that 18.2% of pet owners discontinued antibiotics early because their pets seemed better (11.6%) or because of the difficulty in administering them (6.6%) [13]. Although the pet owners in our study reported discontinuing antibiotics early and not completing the full course of antibiotics for their pets, past studies have suggested that pet owners generally place a high level of trust in their veterinarians’ recommendations [13,14,15]. Therefore, veterinarians can leverage this trust during clinic consultations with pet owners to emphasize the correct dose, frequency, mode and duration when prescribing antibiotics. Effective communication and shared decision-making can also help to identify and address pet owners’ concerns regarding antibiotic administration and ensure better antibiotic compliance.

We observed that pet owners’ knowledge of AMR occurrence in pets was inadequate, as corroborated by numerous studies conducted in Western countries [13,14,15,16,17,18,19]. In our study, nearly two in three pet owners (67%) had poor knowledge of AMR in pets. Of concern, fewer pet owners believed that the occurrence of AMR in pets could affect their health or their family’s health, compared to the occurrence of AMR in humans (47.2% vs. 69.5%; *p* < 0.001). This limited knowledge of the potential AMR transmission risk between pet owners and their pets in our study is alarming, especially when it has been reported that pet owners regard their pets as “my fur babies” [12]. Understanding the close bonds and deeply affectionate relationships between pet owners and their pets that shape the behaviors that promote the interspecies transmission of AMR is crucial. Likewise, owners’ unconditional love for their pets can be leveraged to address the owners’ knowledge and practice gaps. While the transmission of AMR can occur both ways, emphasizing the possibility that pets can acquire resistance genes from their owners, rather than highlighting the contrary, can create impactful messages that tug at pet owners’ heartstrings.

The strength of our study lies in using the established WHO multi-country survey questions [37] to assess pet owners’ knowledge of antibiotic use and AMR for themselves and their pets. To our knowledge, this is the only study to date that compares pet owners’ use of antibiotics for themselves and for their pet(s). Our findings will add to the literature on the importance of a One Health approach to increase the understanding of antibiotic use and AMR in humans and animals.

We acknowledge that our study could be limited by social desirability bias. However, as the survey was self-administered, online and anonymous, we believe that any bias is likely to be minimal. The respondents in our study may not be representative of pet owners in Singapore, as we purposively sampled veterinary clinics by practice types and locations across the country. Soliciting pet owners in veterinary clinics to participate in our survey may have led to increased enrollment of owners with sick pets having prior antibiotic use experience. This may have created biased results that limit the generalizability of our study outcomes [38,39]. Nevertheless, the profile of our survey participants was similar to other pet-owner studies from Western countries [13,14,19,40,41] that employed social media, third-party survey platforms, in-clinic/hospital clients or a combination of these approaches. Our findings may therefore be applied to other countries.

## 4. Methods

### 4.1. Study Design and Settings

From 1 March 2023 to 31 December 2023, a cross-sectional study was conducted at 16 veterinary clinics located across Singapore. The sampling frame consisted of approximately 100 licensed veterinary clinics under the Animal and Veterinary Service (AVS) in Singapore. We first stratified the clinics by their location (north, south, east, west and central) and then by the size of the practice (solo/small group: 2 or fewer clinics under the same name; large chain: more than 2 clinics under the same name) to account for the diverse organizational practices relating to pet management. Eligible pet owners (aged 21 years and above) of dogs and/or cats were recruited consecutively from 3 to 4 clinics purposively sampled from each of the 5 zones to ensure a satisfactory representation of solo and large veterinary general practices across Singapore. As they brought their pets in for consultation at the clinics, the participants were invited to complete a self-administered survey hosted on an online platform. The study was approved by the Domain Specific Review Board, National Healthcare Group, Singapore (reference number: 2021/00769).

### 4.2. Survey Instrument

The anonymous questionnaire included questions on socio-demographics (age in years, gender [male/female], ethnicity [Chinese/non-Chinese], marital status [yes/no], education [post-secondary and below as lower educated/diploma and higher as higher educated]); total number of pets ever owned; type of pet(s) owned; pet ownership duration (years), 3 items on the knowledge of antibiotic use (Yes/No/Don’t know or True/False/Don’t know) for self and 8 items on the knowledge of AMR occurrence (True/False/Don’t know) in humans from the WHO’s Antibiotic Resistance Multi-country Public Awareness Survey [37], 3 items on the knowledge of antibiotic use for their pets and 9 items on the knowledge of AMR occurrence in their pets, adapted accordingly.

The pet owners were determined to have poor knowledge of antibiotic use if they incorrectly answered any of the 3 items on antibiotic use for themselves or their pets. For AMR knowledge in humans and pets, the total composite score was 8 and 9, respectively, based on all correct responses, and pet owners who scored less than the 75th percentile (i.e., below 5) were deemed to have poor knowledge of AMR occurrence. Additionally, 6 proxy statements on the inappropriate use of antibiotics for themselves and their pets were based on the U.S. Centers for Disease Control and Prevention’s advisory on appropriate antibiotic use [42]. The pet owners who agreed or strongly agreed with any of the 6 proxy statements referring to themselves or their pet were defined as having used antibiotics inappropriately for themselves or their pet. The pet owners of only dogs and the pet owners of both dogs and cats were categorized as “dog owners”. The pet owners of only cats were categorized as “non-dog owners” (Appendix A).

### 4.3. Data Analysis

The proportions and medians (interquartile range, IQR) were used to describe the categorical and continuous variables, respectively. We used McNemar’s Chi-square test to make within-subject comparisons of the main outcome variables between self and their pets. Pearson’s Chi-square test was used to identify associations between the type of pet owned (dog owner/non-dog owner) and individual statements (as well as overall) for the knowledge of antibiotic use, AMR, and inappropriate antibiotic use. Statistical significance was defined as *p* < 0.05. The statistical analyses were conducted using Stata version 14.0 (StataCorp LLC, College Station, TX, USA).

## 5. Conclusions

Pet owners have significantly poorer antibiotic use knowledge and antibiotic practices for their pets than for themselves. Compared to dog owners, non-dog owners have poorer knowledge of antibiotic use. Educational initiatives addressing pet owners’ knowledge deficits may improve antibiotic practices for their pets and themselves, emphasizing the importance of cross-sectoral One Health planning.

## Figures and Tables

**Figure 1 antibiotics-14-00158-f001:**
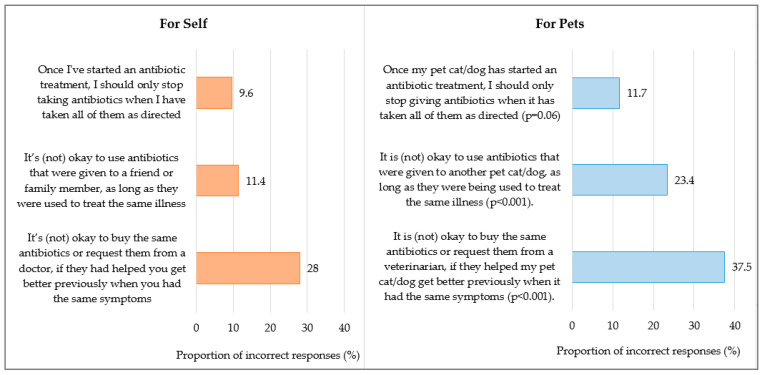
Knowledge of antibiotic use for self and for pets among 1080 pet owners of dogs and/or cats. *p* value based on McNemar Chi-square test.

**Figure 2 antibiotics-14-00158-f002:**
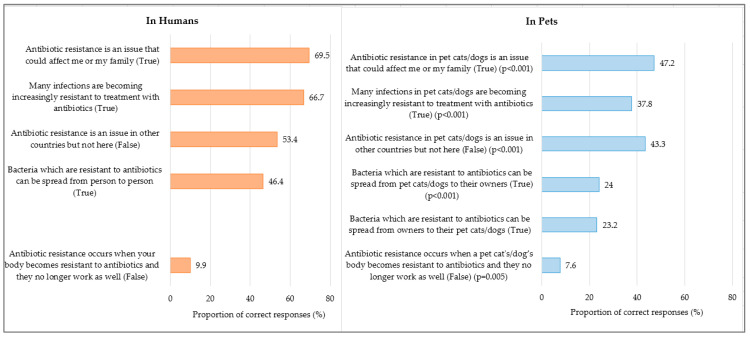
Knowledge of AMR in humans and in pets among 1080 pet owners of dogs and/or cats. *p* value based on McNemar Chi-square test.

**Figure 3 antibiotics-14-00158-f003:**
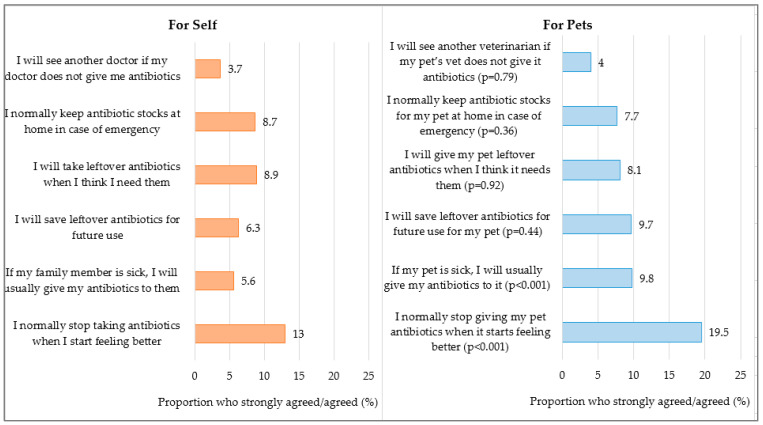
Practice of inappropriate antibiotic use for self and for pets among 1080 pet owners of dogs and/or cats. *p* value based on McNemar Chi-square test.

**Table 1 antibiotics-14-00158-t001:** Characteristics of pet owners of dogs and/or cats–overall and by type of pet owned (N = 1080).

Demographics	Total (N = 1080)	Own Dogs (N = 754)	Do Not Own Dogs (N = 326)
**Age group, in years, N(%)**			
21–34	399 (36.9)	262 (34.8)	137 (42.0)
35–49	382 (35.4)	264 (35.0)	118 (36.2)
50 and above	299 (27.7)	228 (30.2)	71 (21.8)
**Gender, N(%)**			
Male	390 (36.1)	280 (37.1)	110 (33.7)
Female	690 (63.9)	474 (62.9)	216 (66.3)
**Ethnicity, N(%)**			
Chinese	798 (73.9)	621 (82.4)	177 (54.3)
Malay	91 (8.4)	3 (0.4)	88 (27.0)
Indian	70 (6.5)	55 (7.3)	15 (4.6)
Others	121 (11.2)	75 (10.0)	46 (14.1)
**Highest education level, N(%)**			
Lower educated (GCE-A level & below)	141 (13.1)	99 (13.1)	42 (12.9)
Higher educated (Diploma & above)	939 (86.9)	655 (86.9)	284 (87.1)
**Marital status, N(%)**			
Married	607 (56.2)	426 (56.5)	181 (55.5)
Never married/widowed/divorced/separated	473 (43.8)	328 (43.5)	145 (44.5)
**Housing type, N(%)**			
1- to 4-room HDB apartments	378 (35.0)	226 (30)	152 (46.6)
5-room HDB; Privately-owned apartments, condominiums, and houses	702 (65.0)	528 (70)	174 (53.4)
**Employment status, N(%)**			
Currently working	825 (76.4)	574 (76.1)	251 (77.0)
Retired/not working	255 (23.6)	180 (23.9)	75 (23.0)
**Occupation, N(%)**			
Not related to human or animal health	833 (77.1)	590 (78.3)	243 (74.5)
Related to human or animal health	247 (22.9)	164 (21.8)	83 (25.5)
**Duration of pet ownership, N(%)**			
≤10 years	605 (56.0)	393 (52.1)	212 (65.0)
>10 years	475 (44.0)	361 (47.9)	114 (35.0)
**Knowledge of antibiotic use for pets, N(%)**			
Not poor	580 (53.7)	434 (57.6)	146 (44.8)
Poor	500 (46.3)	320 (42.4)	180 (55.2)
**Knowledge of AMR in pets, N(%)**			
Not poor	355 (32.9)	258 (34.2)	97 (29.8)
Poor	725 (67.1)	496 (65.8)	229 (70.3)
**Use of antibiotics for pets, N(%)**			
Appropriate	714 (66.1)	493 (65.4)	221 (67.8)
Inappropriate	366 (33.9)	261 (34.6)	105 (32.2)
**Knowledge of antibiotic use for self, N(%)**			
Not poor	722 (66.9)	521 (69.1)	201 (61.7)
Poor	358 (33.2)	233 (30.9)	125 (38.3)
**Knowledge of AMR in humans, N(%)**			
Not poor	379 (35.1)	269 (35.7)	110 (33.7)
Poor	701 (64.9)	485 (64.3)	216 (66.3)
**Use of antibiotics for self, N(%)**			
Appropriate	826 (76.5)	579 (76.8)	247 (75.8)
Inappropriate	254 (23.5)	175 (23.2)	79 (24.2)

HDB—Housing Development Board (government-subsidized housing).

## Data Availability

The datasets from the current study are available from the corresponding author upon reasonable request.

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
