# Peer review of "Pet Owners’ Knowledge of Antibiotic Use and Antimicrobial Resistance and Their Antibiotic Practices: Comparison Between Contexts of Self and Pet"

_antibiotics, 2025, doi:10.3390/antibiotics14020158_

Round 1
Reviewer 1 Report
Comments and Suggestions for Authors
The paper “Comparison of Pet Owners’ Knowledge and Practice of Antibiotic Use and Antimicrobial Resistance for Self and their Pets” is a survey with the scope to underline the knowledge of pet owners towards antibiotic correct use and antibiotic resistance.
The obtained results can be of interest, unfortunately the paper should be improved before to be published. In general, The English text needs to be greatly improved as in this version in some cases it is difficult to understand what the Authors mean. Please find only few examples:
- “antibiotic practices for themselves and their pets and by pet type (dog owners vs non-dog owners)”; it is difficult to understand what the authors mean with “pet type” even if an explication is given in brackets a different way to indicate the two population should be addressed, as well as in the Title the sentence “for self and their pets” should be changed;
- “only one-third (35%, 114/326) of non-dog owners had >10 years of pet ownership experience”
It’s not clear what the authors mean.
- “There was no significant difference between pet owners’ overall understanding of
AMR issue in humans and in pets”; between pet owners and who?
- “More dog owners were cognizant of infections becoming increasing resistant to treatment by antibiotics, compared to non-dog owners”;
- “Likewise, whilst only 5.6% of pet owners reportedly shared their antibiotics 119 with a sick family member, nearly twice (9.8%) as many pet owners gave their antibiotics 120 to their sick pets (P<0.001)”
- “Pet owners’ heightened sense of responsibility for their vulnerable pets, coupled with inexperience in managing pet illnesses, may explain their poor antibiotic practices”
- “Of concern, fewer pet owners believed AMR in pets can affect their health or their family’s health 156 compared to AMR in the humans affecting their health or family’s health (47.2% vs 69.5%, 157 P<0.001)”
- “Understanding the close bond and deeply affectionate relation-160 ships between pet owners and their pets which shape the behaviours promoting interspe-161 cies transmission is key to reducing AMR.”
The Title should be changed.
As regards the recruitment of responders, it is not clear where the population of non-owners has been enrolled (maybe not in veterinary clinics).
Comments on the Quality of English Language
The English text needs to be greatly improved as in this version in some cases it is difficult to understand what the Authors mean.
Author Response
Comments and Suggestions for Authors
The paper “Comparison of Pet Owners’ Knowledge and Practice of Antibiotic Use and Antimicrobial Resistance for Self and their Pets” is a survey with the scope to underline the knowledge of pet owners towards antibiotic correct use and antibiotic resistance.
The obtained results can be of interest, unfortunately the paper should be improved before to be published. In general, The English text needs to be greatly improved as in this version in some cases it is difficult to understand what the Authors mean. Please find only few examples:
- “antibiotic practices for themselves and their pets and by pet type (dog owners vs non-dog owners)”; it is difficult to understand what the authors mean with “pet type” even if an explication is given in brackets a different way to indicate the two population should be addressed, as well as in the Title the sentence “for self and their pets” should be changed;
Response: Thank you for reviewing our manuscript. Our study aims to compare pet owners’ knowledge of antibiotics use and AMR, as well as their antibiotic use practices for themselves and for their pets respectively. For this first comparison, there were separate sets of questions assessing pet owners’ knowledge of antibiotic use and AMR, and antibiotic use practices, for themselves and their pets (please find survey questionnaire appended in the supplementary materials for your reference). So, here we are comparing knowledge and practices for pet owners themselves and their pets in the same group of pet owners. Hence, we used McNemar Chi square test for paired comparison in assessing statistical significance in the differences observed.
Additionally, we also compared pet owners who owned dogs with pet owners who did not own dogs, by stratifying the entire pet owner cohort by the type of pets they owned (dog owner/non-dog owner). Then, we used Pearson’s Chi square test to assess differences in knowledge of antibiotic use and AMR, as well as practices in antibiotic use, between the 2 groups of pet owners (dog owners vs. non-dog owners).
Hopefully, the above explanation dispels any confusion caused while reviewing our paper. Thank you.
- “only one-third (35%, 114/326) of non-dog owners had >10 years of pet ownership experience”
It’s not clear what the authors mean.
Response: Here we compare the duration of pet ownership (>10 years of pet ownership was defined as “experienced pet owner”) between dog owners and non-dog owners. Thank you.
- “There was no significant difference between pet owners’ overall understanding of AMR issue in humans and in pets”; between pet owners and who?
Response: As mentioned earlier, we performed a paired test comparison in the same group (entire cohort) of pet owners using 2 different sets of questions to assess AMR knowledge for AMR occurrence in themselves (humans) and for their pets. We found that there was no significant difference in pet owners’ overall AMR knowledge (defined according to the definition stated in the methods section) for AMR occurrence in themselves (humans) and their pets. Thank you.
- “More dog owners were cognizant of infections becoming increasing resistant to treatment by antibiotics, compared to non-dog owners”;
- “Likewise, whilst only 5.6% of pet owners reportedly shared their antibiotics 119 with a sick family member, nearly twice (9.8%) as many pet owners gave their antibiotics 120 to their sick pets (P<0.001)”
- “Pet owners’ heightened sense of responsibility for their vulnerable pets, coupled with inexperience in managing pet illnesses, may explain their poor antibiotic practices”
- “Of concern, fewer pet owners believed AMR in pets can affect their health or their family’s health 156 compared to AMR in the humans affecting their health or family’s health (47.2% vs 69.5%, 157 P<0.001)”
- “Understanding the close bond and deeply affectionate relation-160 ships between pet owners and their pets which shape the behaviours promoting interspe-161 cies transmission is key to reducing AMR.”
Response: We have expanded on the Introduction section to provide a more detailed description of the research gaps and provide a stronger basis for the research focus on Singapore. We have checked for grammatical errors and edited the text, where appropriate, to improve the overall readability. Furthermore, we have amended the Discussion section to improve on the interpretation of our study findings whist citing relevant literature. Thank you.
The Title should be changed.
Response: Noted with thanks. We have revised the study title as advised.
As regards the recruitment of responders, it is not clear where the population of non-owners has been enrolled (maybe not in veterinary clinics).
Response: As mentioned in the Methods section, only eligible pet owners (aged 21 years and above) of dogs and/or cats were recruited when they attended veterinary clinics for consultation. They were categorised into 2 groups – dog owners and non-dog owners based on the type of pet owned. Thank you.
Comments on the Quality of English Language
The English text needs to be greatly improved as in this version in some cases it is difficult to understand what the Authors mean.
Response: We thank the Reviewer for his/her feedback. We have carefully combed through and edited our paper, according to the Reviewer’s feedback. We hope that our revised manuscript has met the standard required by the journal. Thank you.
Submission Date
13 December 2024
Date of this review
28 Dec 2024 13:28:49
Reviewer 2 Report
Comments and Suggestions for Authors
The manuscript is interesting and has relevant results that have a great social impact that could explain in some way the current problem of antimicrobial resistance, however I have some observations:
What was the importance of knowing some socioeconomic aspects of the people surveyed? Was there no correlation of this data with the questions asked?
A question that could have been added is whether the veterinarian provided all the information on the treatments and whether or not antimicrobials were necessary and what would happen if the treatment was not completed, something like that.
They could improve the quality of the graphics or present the results in a different way; the font is very small. They could also include the corresponding figures after each result where they are referred to.
Author Response
Comments and Suggestions for Authors
The manuscript is interesting and has relevant results that have a great social impact that could explain in some way the current problem of antimicrobial resistance, however I have some observations:
What was the importance of knowing some socioeconomic aspects of the people surveyed? Was there no correlation of this data with the questions asked?
Response: We collected socio-demographic data from pet owners surveyed in-person from purposively sampled veterinary clinics across Singapore to facilitate comparison of pet owners’ characteristics with other pet owner studies conducted in Western countries which used social media and third party survey platforms to recruit participants. Also, such data will be useful while planning and designing educational interventions to target specific groups of pet owners. We believe that pet ownership reflects some degree of affluence in Singapore as there are restrictions on the number of pets and type of pet breed (especially for dogs) you can own while living in HDB blocks (government subsidized housing – where 80% of the resident population live) whereas such restrictions do not apply for private housing. In our study we found that a higher proportion of dog owners lived in larger housing.
We found no correlation with the main outcome variables. None of the socio-economic characteristics were found to be significant when we looked at the factors associated with inappropriate use of antibiotics in pets (this is part of a separate paper which has been accepted for publication). Thank you.
A question that could have been added is whether the veterinarian provided all the information on the treatments and whether or not antimicrobials were necessary and what would happen if the treatment was not completed, something like that.
Response: Our paper is a descriptive study seeking to compare the knowledge of antibiotic use and AMR and antibiotic use practices for themselves and their pets. We did not look into the determinants of inappropriate antibiotic use for pets in the present study. However, in the survey, we did enquire pet owners if they had ever gotten advice from their veterinarians on how to administer antibiotics for their pets and also included questions on shared decision making during their last veterinary consultation. The results from these questions are part of a separate paper which is under review for publication. Thank you for this valuable insight.
They could improve the quality of the graphics or present the results in a different way; the font is very small. They could also include the corresponding figures after each result where they are referred to.
Response: Noted. We have updated the figures to improve the quality of the graphics. We have also split the larger figures to include the most important information only and created a new table with the remaining data which has been appended in supplementary materials. Thank you for your suggestion.
Submission Date
13 December 2024
Date of this review
15 Jan 2025 04:25:21
Reviewer 3 Report
Comments and Suggestions for Authors
Dear Authors,
The introduction establishes the context of antimicrobial resistance (AMR) and emphasizes the relevance of a One Health approach. It includes global perspectives and highlights the gap in studies focusing on Asia. However, it lacks more nuanced discussion on specific behavioral interventions or prior research directly comparing antibiotic practices between pet owners and their pets.
The methodology is well-detailed, including the sampling process, survey instruments, and statistical analysis. However, there is limited explanation about potential biases from participant recruitment through veterinary clinics.
Results are presented with appropriate tables and figures, making comparisons between pet owners’ practices for themselves and their pets clear. However, the discussion of the results occasionally overlaps with interpretation, which could be better distinguished.
The conclusions align with the results, highlighting poor knowledge and practices for pets compared to humans. The study's implications for educational initiatives and One Health planning are well-founded.
The English language is clear and professional, with only minor grammatical and stylistic issues.
The references are recent and pertinent, particularly those on AMR and One Health. However, adding more studies from Asian contexts could strengthen the regional focus.
Comments and Suggestions for Authors:
1. Consider expanding the introduction to include more region-specific studies to provide a stronger foundation for the research focus on Singapore.
2. Clarify the limitations related to recruiting participants through veterinary clinics, as this may introduce selection bias.
3. Separate result presentation from interpretation in the discussion section for better clarity.
4. Minor grammatical refinements and editing would enhance the overall readability.
5. Keywords: do not repeat words that are in the title. Please change.
6. Evaluate the quality of the figures. There is a lot of information in the figures, which makes them difficult to read. Try to summarize the parts written in the figures as much as possible and, if necessary, leave the most important data and create new tables as supplementary documents.
Best regards!
Author Response
Comments and Suggestions for Authors
Dear Authors,
The introduction establishes the context of antimicrobial resistance (AMR) and emphasizes the relevance of a One Health approach. It includes global perspectives and highlights the gap in studies focusing on Asia. However, it lacks more nuanced discussion on specific behavioral interventions or prior research directly comparing antibiotic practices between pet owners and their pets.
The methodology is well-detailed, including the sampling process, survey instruments, and statistical analysis. However, there is limited explanation about potential biases from participant recruitment through veterinary clinics.
Results are presented with appropriate tables and figures, making comparisons between pet owners’ practices for themselves and their pets clear. However, the discussion of the results occasionally overlaps with interpretation, which could be better distinguished.
The conclusions align with the results, highlighting poor knowledge and practices for pets compared to humans. The study's implications for educational initiatives and One Health planning are well-founded.
The English language is clear and professional, with only minor grammatical and stylistic issues.
The references are recent and pertinent, particularly those on AMR and One Health. However, adding more studies from Asian contexts could strengthen the regional focus.
Comments and Suggestions for Authors:
- Consider expanding the introduction to include more region-specific studies to provide a stronger foundation for the research focus on Singapore.
Response: Noted with thanks. To date there is no data comparing antibiotic use practices for themselves and their pets. We have expanded on the introduction to highlight the specific research gaps in the regional context thereby strengthening the rationale for our study.
- Clarify the limitations related to recruiting participants through veterinary clinics, as this may introduce selection bias.
Response: Thank you for pointing this. We have now incorporated this under study limitations in the Discussion section.
- Separate result presentation from interpretation in the discussion section for better clarity.
Response: We have revised the discussion section to better distinguish our study findings whilst citing relevant literature. Thank you for pointing this.
- Minor grammatical refinements and editing would enhance the overall readability.
Response: We have checked for grammatical errors and edited where appropriate to improve overall readability. Thank you.
- Keywords: do not repeat words that are in the title. Please change.
Response: Noted with thanks. We have amended the keywords list as advised.
- Evaluate the quality of the figures. There is a lot of information in the figures, which makes them difficult to read. Try to summarize the parts written in the figures as much as possible and, if necessary, leave the most important data and create new tables as supplementary documents.
Response: Thanks for the suggestion. We have updated the figures to improve the quality. Additionally, we have amended the figures as advised to include only relevant and important information described in the methods and discussion section. Remaining data is presented as a new table appended under supplementary materials.
Best regards!
Round 2
Reviewer 1 Report
Comments and Suggestions for Authors
Dear authors,
the paper has been widely improved. However, some minor critical issues should be addressed.
Title - I suggest changing the title as: "Pet Owners’ Knowledge of Antibiotic Use, Antimicrobial Resistance, and Antibiotic Practices: Comparison between 3 Contexts of Self and Pet."
Abstract
Line 20: please change "knowledge of antibiotic use", to "knowledge on antibiotic use"
Line 28: it is not clear what the authors mean witih "Pet owned". Is it "pet-owners". If yes please modify.
Results
Lines 87 and 90: please change "larger" to "large".
Lines 148, 151: please remove "reportedly". The sentence "when they them-148 selves started feeling better" should be improved.
Lines 152-153: the sentence "nearly twice (9.8%) as many pet owners gave their antibiotics 152 to their sick pets" is not clear. Please rephrase.
Line 184: please change "prescribing" to "prescription"
Line 196: what do the authors mean with "feeding concerns". It is not clear.
Methods
line 270: to improve the clarity of the study desing, a specification on what the authors mean with "non-dog owner" could be addressed.
Round 2
Reviewer 1
Open Review
(x) I would not like to sign my review report
( ) I would like to sign my review report
Quality of English Language
(x) The English is fine and does not require any improvement.
( ) The English could be improved to more clearly express the research.
Yes |
Can be improved |
Must be improved |
Not applicable |
|
Does the introduction provide sufficient background and include all relevant references? |
(x) |
( ) |
( ) |
( ) |
Is the research design appropriate? |
(x) |
( ) |
( ) |
( ) |
Are the methods adequately described? |
( ) |
(x) |
( ) |
( ) |
Are the results clearly presented? |
(x) |
( ) |
( ) |
( ) |
Are the conclusions supported by the results? |
(x) |
( ) |
( ) |
( ) |
Comments and Suggestions for Authors
Dear authors,
the paper has been widely improved. However, some minor critical issues should be addressed.
Response: Thank you for reviewing our manuscript.
Title - I suggest changing the title as: "Pet Owners’ Knowledge of Antibiotic Use, Antimicrobial Resistance, and Antibiotic Practices: Comparison between 3 Contexts of Self and Pet."
Response: Thank you for your suggestion. Our study refers to comparison of 2 contexts – first context is pet owners’ knowledge of antibiotic use and AMR and antibiotic practice for themselves, the second context is pet owners’ knowledge of antibiotic use and AMR and antibiotic practice for their pet. Hope this clarifies. As such, we will keep to our proposed title.
Abstract
Line 20: please change "knowledge of antibiotic use", to "knowledge on antibiotic use"
Response: Amended with thanks.
Line 28: it is not clear what the authors mean witih "Pet owned". Is it "pet-owners". If yes please modify.
Response: We have amended the statement to: “… but knowledge of AMR occurrence and antibiotic practice did not differ between dog and non-dog owners”. Thank you.
Results
Lines 87 and 90: please change "larger" to "large".
Response: Changed. Thanks.
Lines 148, 151: please remove "reportedly". The sentence "when they them-148 selves started feeling better" should be improved.
Response: Removed “reportedly”. Amended sentence to: “Whilst 13% of pet owners stopped taking antibiotics when they started feeling better, 19.5% of them stopped giving their pets antibiotics when their pets started feeling better (P<0.001).”
Lines 152-153: the sentence "nearly twice (9.8%) as many pet owners gave their antibiotics 152 to their sick pets" is not clear. Please rephrase.
Response: Rephrased as follows: ”Only 5.6% of pet owners shared their antibiotics with a sick family member, but 9.8% of them gave their antibiotics to their sick pets (P<0.001).”
Line 184: please change "prescribing" to "prescription"
Response: Noted with thanks. Here, we mean the act of ordering antibiotics (prescribing), not the written order for antibiotics (prescription). As such, we have kept “prescribing” in this sentence.
Line 196: what do the authors mean with "feeding concerns". It is not clear.
Response: For better clarity, we have amended the statement to: “Effective communication and shared decision-making can also help to identify and address pet owners’ concerns regarding antibiotic administration and ensure better antibiotic compliance.” Thanks.
Methods
line 270: to improve the clarity of the study desing, a specification on what the authors mean with "non-dog owner" could be addressed.
Response: Thank you for the suggestion. We included the following as the last sentence of the “Survey Instrument” section: ‘Pet owners of only dogs and pet owners of both dogs and cats were categorised as “dog owners”. Pet owners of only cats were categorised as “non-dog owners”.’
Submission Date
13 December 2024
Date of this review
27 Jan 2025 13:11:42
Reviewer 3 Report
Comments and Suggestions for Authors
Congratulations to the authors for knowing how to understand constructive criticism to improve the article. Thank you for considering my opinions and making necessary corrections. The article is now ready for publication.
Round 2
Reviewer 3
Open Review
( ) I would not like to sign my review report
(x) I would like to sign my review report
Quality of English Language
(x) The English is fine and does not require any improvement.
( ) The English could be improved to more clearly express the research.
Yes |
Can be improved |
Must be improved |
Not applicable |
|
Does the introduction provide sufficient background and include all relevant references? |
(x) |
( ) |
( ) |
( ) |
Is the research design appropriate? |
(x) |
( ) |
( ) |
( ) |
Are the methods adequately described? |
(x) |
( ) |
( ) |
( ) |
Are the results clearly presented? |
( ) |
( ) |
( ) |
( ) |
Are the conclusions supported by the results? |
(x) |
( ) |
( ) |
( ) |
Comments and Suggestions for Authors
Congratulations to the authors for knowing how to understand constructive criticism to improve the article. Thank you for considering my opinions and making necessary corrections. The article is now ready for publication.
Response:
Thank you for reviewing our manuscript. Your insightful suggestions have really helped to improve it.
Submission Date
13 December 2024
Date of this review
22 Jan 2025 11:34:17